# PKM2 Modulation in Head and Neck Squamous Cell Carcinoma

**DOI:** 10.3390/ijms23020775

**Published:** 2022-01-11

**Authors:** Verena Boschert, Jonas Teusch, Urs D. A. Müller-Richter, Roman C. Brands, Stefan Hartmann

**Affiliations:** 1Department of Oral and Maxillofacial Plastic Surgery, University Hospital Würzburg, D-97070 Würzburg, Germany; teusch.jon@googlemail.com (J.T.); Mueller_U2@ukw.de (U.D.A.M.-R.); Brands_R@ukw.de (R.C.B.); Hartmann_S2@ukw.de (S.H.); 2Comprehensive Cancer Center Mainfranken, University Hospital Würzburg, D-97070 Würzburg, Germany; 3Bavarian Cancer Research Center (BZKF), D-91054 Erlangen, Germany

**Keywords:** HNSCC, head and neck cancer, cancer metabolism, glycolysis, PKM2, Warburg effect, CD44, Compound 3k, DASA-58, AMPK, TXNIP

## Abstract

The enzyme pyruvate kinase M2 (PKM2) plays a major role in the switch of tumor cells from oxidative phosphorylation to aerobic glycolysis, one of the hallmarks of cancer. Different allosteric inhibitors or activators and several posttranslational modifications regulate its activity. Head and neck squamous cell carcinoma (HNSCC) is a common disease with a high rate of recurrence. To find out more about PKM2 and its modulation in HNSCC, we examined a panel of HNSCC cells using real-time cell metabolic analysis and Western blotting with an emphasis on phosphorylation variant Tyr105 and two reagents known to impair PKM2 activity. Our results show that in HNSCC, PKM2 is commonly phosphorylated at Tyrosine 105. Its levels depended on tyrosine kinase activity, emphasizing the importance of growth factors such as EGF (epidermal growth factor) on HNSCC metabolism. Furthermore, its correlation with the expression of CD44 indicates a role in cancer stemness. Cells generally reacted with higher glycolysis to PKM2 activator DASA-58 and lower glycolysis to PKM2 inhibitor Compound 3k, but some were more susceptible to activation and others to inhibition. Our findings emphasize the need to further investigate the role of PKM2 in HNSCC, as it could aid understanding and treatment of the disease.

## 1. Introduction

Tumor cells rely on glycolysis for their growth, channeling high amounts of glucose through this pathway without subsequent oxidative phosphorylation (OxPhos) [1,2]. Instead, the generated pyruvate is converted to lactate. This phenomenon is called the Warburg effect, because it was first described by Otto Warburg in 1926 [3]. Why this aerobic glycolysis (normally cells employ this metabolic route only when oxygen is limited) is preferred by tumor cells is still a matter of debate [4]. The reason could be that it helps cancer cells to proliferate by providing energy more quickly and delivering precursors for several synthesis pathways in adequate amounts. Other explanations given for the preference of cancer cells for aerobic glycolysis are the modulation of reactive oxygen species or chromatin state. Furthermore, this switch in carbon metabolism could be favorable for the establishment of a microenvironment beneficial for tumor growth.

One of the key steps in glycolysis is the irreversible conversion of phosphoenolpyruvate (PEP) to pyruvate and ATP. This reaction is catalyzed by the enzyme pyruvate kinase (PK). Initially, two isoforms of this enzyme were described; one has been called PKL, as it was isolated from the liver, whereas the other one was labeled PKM because it was found in muscle [5]. Later, it was discovered that there are two differently spliced forms of PKM: PKM1 and PKM2 [6,7]. Another isoform, called PKR, originates from the same gene as PKL and is only expressed in erythrocytes [8].

PKM2 can be found in various tissues and organs of the body; it is also the only PK variant present in embryonic stages [9]. Commonly, PKM2 is more abundant in highly proliferating cells and is overexpressed in several types of cancer [9]. In contrast to PKM1, PKM2 can become allosterically activated or inhibited by modifying its quaternary structure [10,11,12]. Furthermore, several post-translational modifications can influence PKM2 activity or its intracellular location [13]. That PKM2 activity can be strictly regulated compared to PKM1 activity might turn it into a more favorable variant for highly proliferating tumor cells [14]. When PKM2 activity is downregulated, intermediates of glycolysis can be channeled into other pathways generating cellular components such as amino acids or nucleotides. Furthermore, PKM2 is able to regulate gene expression. By activating transcription factors Hif-1α [15,16] and cMyc [17,18], several glycolytic enzymes and the expression of PKM2 itself can be upregulated. Thereby, PKM2 actively promotes the metabolic switch in tumor cells towards aerobic glycolysis.

Head and neck cancer is a highly common disease; in 2018 it accounted for 835,000 new cases of cancer worldwide [19]. About 90% of head and neck cancer cases are reported to be of the squamous cell carcinoma type [20], characterized by an abnormal and quick growth of keratinocytes in the epidermis of the mouth, tongue, nasal cavity, nasopharynx and throat. Consuming tobacco, alcohol abuse, poor oral hygiene and human papillomavirus (HPV) infection are the main risk factors for developing head and neck squamous cell carcinoma (HNSCC). Metabolomic studies indicate that in HNSCC, aerobic glycolysis is the major route of carbon metabolism [21]. This is reflected in the application of glucose analogues such as fluorodeoxyglucose (FDG) in positron emission tomography (PET) for diagnosis and response assessment of HNSCC [22]. The high uptake of glucose by HNSCC cells makes it possible to detect the tumors in patients. Although already proven very helpful for diagnosis, interventions addressing HNSCC metabolism have not been approved to date. Therapy comprises mainly surgical interventions combined with irradiation and chemotherapy [23]. A recently approved approach showing considerable success is the treatment with immune checkpoint inhibitors (ICIs) targeting the PD1/PD-L1 axis. Unfortunately, the overall response rates (ORR) of this treatment are not satisfactory [24].

There have not been many mechanistic investigations into the role of PKM2 in HNSCC to date. However, there is some evidence from a recent meta-analysis that several markers of glycolysis (e.g., GLUT1, MCT4, HK2 and PKM2) might play an important role in HNSCC and are associated with an unfavorable outcome [25]. Therefore, we wanted to look for the expression levels of PKM2 and one of its phosphorylated variants, P-PKM2 Tyr105, in a set of established HNSCC cell lines. PKM2 molecules phosphorylated at this residue do not exist in their active tetrameric form due to loss of binding to their allosteric activator, fructose-1, 6-bisphosphate (FBP) [26]. Furthermore, it was shown that phosphorylation at Tyr105 is able to induce cancer stem cell-like properties in breast cancer cell lines [27]. Hence, we investigated if the amount of P-PKM2 Tyr105 detected in the HNSCC cell lines correlates with the expression of cancer stem cell marker gene CD44.

Several growth factor receptors with tyrosine kinase activity (e.g., EGFR, FGFR and others) were found to be responsible for the phosphorylation at residue 105 of PKM2 [26,27]. Their ability to perform this phosphorylation was found to be crucial for a metabolic state showing high levels of aerobic glycolysis and low levels of OxPhos. In this context, the epidermal growth factor (EGF) has a pivotal role as a known contributor of the glycolytic switch in HNSCC as well as a major driver of HNSCC initiation and progression [28]. With our study, we aimed to clarify the role of EGF, HGF and FGF as drivers of the Warburg effect in HNSCC via the promotion of PKM2 phosphorylation.

Additionally, we wanted to investigate how HNSCC tumor cells react to PKM2 activator DASA-58 and PKM2 inhibitor Compound 3k in terms of cell viability and carbon metabolism. DASA-58 treatment is reported to lead to PKM2 activation by inducing tetramerization of the kinase and suppressing lactate secretion and tumorigenesis [29]. PKM2 inhibitors such as Compound 3k, on the other hand, showed cytotoxic activity and led to higher oxygen consumption in ovarian cells [30].

## 2. Results

To investigate PKM2 in head and neck squamous cell carcinoma (HNSCC) we chose six cell lines from different origins: SCC-9 (tongue) [31], FaDu (laryngopharynx) [32], Detroit 562 (pharynx-derived metastasis from pleural effusion) [33], HN (soft palate-derived metastasis from a cervical lymph node) [34], BHY (alveolar ridge) [34] and SCC-154 (tongue, HPV-positive) [35].

At first, we looked at the protein levels of PKM2 expressed in the HNSCC cell lines using Western blot with an antibody specifically recognizing PKM2 (Figure 1a,b on the left). As positive control, we used the cervical carcinoma cell line HeLa, which is published to express PKM2 [36]. In all six cell lines, PKM2 could be detected with intensities comparable to HeLa or even at slightly higher levels (Detroit 562 4.2 ± 3.3 fold, SCC-9 2.8 ± 2.4 fold). Next, we repeated the experiment using an antibody specific for a PKM2 variant phosphorylated at Tyrosine 105 (P-PKM2 Tyr105). Detroit-562 cells showed the highest levels (10.56 ± 6.2 fold more compared to HeLa) followed by BHY (3.8 ± 0.3 fold) and SCC-9 (3.3 ± 2.1 fold) cell lines (Figure 1a,b on the right). It is known that Tyr105 phosphorylated PKM2 activates YES-associated protein (YAP) signaling, which in turn induces cancer stem-like properties [27]. Therefore, we looked at the expression levels of cancer stem-like marker gene and YAP target CD44 using data from an mRNA sequencing we previously performed with three of our HNSCC cell lines [37]. Indeed, the results for CD44 expression correlated with the levels of the PKM2 variant, with Detroit 562 cells showing the highest levels (Figure 1c).

It was published that the level of phosphorylated PKM2 variant Tyr105 is dependent on the activity of growth factor receptors [26,27], so we chose the HNSCC cell line with the highest variant level, Detroit-562, for investigating the influence of the growth factors epidermal growth factor (EGF), fibroblast growth factor 2 (FGF2) and hepatocyte growth factor (HGF) on phosphorylation. Interestingly, stimulation with EGF significantly increased the level of P-PKM2 Tyr105, whereas adding the EGF-receptor (EGFR) and human epidermal growth factor receptor (HER2) specific tyrosine-kinase-inhibitor (TKI) lapatinib significantly decreased levels of the phosphorylated variant, even below unstimulated levels and without additional EGF stimulation (Figure 2a,b on the right). Levels of total PKM2 protein were not affected (Figure 2a,b on the left). Stimulation with FGF2 and HGF showed no consistently higher levels of PKM2 or its phosphorylated variant; only a tendency to higher levels with FGF2 stimulation could be observed (Appendix A). Furthermore, adding specific TKIs AZD4547 (for FGF-receptor inhibition) or foretinib (for inhibition of HGF receptor Met) alone or in combination with the growth factors resulted only in a tendency to smaller variant levels, but no significant results (Appendix A).

In recent years, several activators or inhibitors with specificity for PKM2 have been developed. We investigated if DASA-58, reported to enhance PKM2 activity, and Compound 3k, a PKM2 inhibitor, showed an impact on the viability of our HNSCC cell lines. Therefore, we tested both substances in a range of concentrations. Cells treated with DASA-58 showed no reaction beyond the effect of its vehicle, DMSO, as exemplarily shown for cell lines SCC-9 and BHY in Figure 3a (black dots). With Compound 3k on the other hand, a clear cytotoxic effect could be observed in all cell lines (Figure 3a, blue squares, Figure 3b). IC_50_-values (inhibitor concentration at 50% viability) ranged between 10.5 ± 5.8 µM for SCC-9 and 23.6 ± 7.7 µM for FaDu (Figure 3b,c).

Next, we tested if DASA-58 and Compound 3k treatment have an impact on the glucose metabolism of HNSCC cell lines. Therefore, we performed glycolytic rate assays, enabling us to measure the extent of glycolysis and OxPhos taking place in the cells under normal culture conditions and under conditions without a functional respiratory chain by injecting electron chain inhibitors. In principle, pH changes of the medium are measured to receive the extracellular acidification rate (ECAR) and changes in oxygen concentration to receive the oxygen consumption rate (OCR). Both rates are used to determine the glycolytic rate (glycoPER) as a measure of lactate secretion.

For a first overview, basal ECAR values obtained after 5 h treatment with 30 µM DASA-58, Compound 3k or vehicle only (Control) were plotted against basal OCR values (Figure 4a). This kind of plot enables one to compare the metabolic state of the cell lines with each other and to observe the effects of treatment. Cells are classified as aerobic when they show low ECAR values combined with high OCR values. The glycolytic type shows high ECAR values combined with low OCR values. Cells showing low ECAR and OCR values are classified as quiescent, whereas cells combining high ECAR and OCR values are classified as energetic. The untreated controls (dots) showed that BHY, Detroit 562 and SCC154 were in a more quiescent state compared to the other three lines (Figure 4a). HN and SCC9 can be classified as energetic, and FaDu as glycolytic. Treatment of FaDu and SCC-9 with Compound 3k changed their classification to aerobic, as OCR became higher and ECAR values lower (black triangles versus light grey dots, upper and lower panel). The other four cell lines showed no strong reactions to the treatment. DASA-58 could not induce changes as pronounced as Compound 3k. Nevertheless, cell lines BHY and SCC154 both exhibited significantly lower OCR values and higher ECAR values (medium green squares versus light green dots, upper and lower panel). Detroit 562 and HN cells showed no strong reaction on treatment with either reagent (blue symbols in upper and lower panel).

The detailed kinetic graphs of FaDu and SCC-9 cells in Figure 4b show that, after 5 h of Compound 3k treatment, cells started into the assay with a lower glycolytic rate (glycoPER) and a higher OCR (data points 1 to 3, Figure 4b, light blue and light red triangles). Furthermore, after Rot/AA injection, cells showed a smaller increase in glycoPER and a smaller decrease in OCR (data points 4 to 6). BHY cells showed no response to Compound 3k (light blue triangles), but when treated with DASA-58 for 5 h, started with a significantly higher glycolytic rate and lower OCR (data points 1 to 3, medium blue and red triangles) and likewise depicted a higher glycolytic rate and lower OCR after Rot/AA injection (data points 4–6). Kinetic graphs after 5 h treatment for the other three cell lines can be found in Appendix A.

Additionally, we performed glycolytic rate assays after 0.5 h and 16 h of DASA-58 and Compound 3k treatment (Appendix A, respectively). For comparing the different time points and treatments, the GlycoPER value at the last measurement before Rot/AA injection was used as a degree for basic glycolysis, whereas the GlycoPER value of the measurement right before 2-DG injection was used as a degree for compensatory glycolysis. FaDu and SCC-9 showed a significantly decreased compensatory glycolysis, with values down to 0.4 fold of vehicle-only-treated control (Figure 5, grey bars). This shows that Compound 3k decreases not only the basal glycolysis under normal culture conditions (Figure 4b and Appendix A), but also the reserve of glycolysis useful in case of special conditions, as for example anaerobic conditions or a switch to aerobic glycolysis in the tumor. DASA-58, on the other hand, enhances this reserve, as can be seen for BHY, SCC154 and HN cell lines (Figure 5, black bars). Basal glycolysis is increased by DASA-58 as well in these lines (Appendix A).

In summary, DASA-58 could induce strong metabolic changes in three out of six HNSCC cell lines, no matter if classified as energetic or quiescent (Table 1). Compound 3k, on the other hand, induced strong metabolic changes in two cell lines, SCC-9 and FaDu, and not in any of the cell lines classified as quiescent (Table 2). Detroit 562, the cell line with the highest level of PKM2 and its Tyr105 variant showed no reaction upon treatment with either reagent (Table 1 and Table 2).

DASA-58 and Compound 3k both have been found to enhance levels of an AMP activated protein kinase (AMPK) variant phosphorylated at Thr172 in breast cancer and ovarian cancer cells respectively [30,38]. This phosphorylation is required for the activation of this kinase with functions in energy homeostasis [39]. Additionally, levels of thioredoxine interacting protein (TXNIP) decreased upon DASA-58 treatment in breast cancer cells [38]. We looked at the levels of both proteins in SCC-9, FaDu and BHY cells after 24 h of DASA-58 and Compound 3k treatment. In FaDu cells, both reagents had no effect on AMPK activation; in SCC-9 cells, a small decrease with DASA-58 treatment was observable (Figure 6). Only in the cell line BHY, which had reacted on DASA-58 with an enhanced glycolytic rate (Figure 4 and Figure 5), levels of the phosphorylated protein increased two fold. Compound 3k treated cells did not show higher amounts of phosphorylated AMPK compared to control (Figure 6). TXNIP levels, on the other hand, strongly decreased in cell lines SCC-9 and FaDu, which had reacted very strongly to the treatment, showing lower glycolysis rates and higher oxygen consumption (Figure 4 and Figure 5). DASA-58 had no effect on TXNIP in both cell lines. In contrast, in BHY cells, Compound 3k treatment showed no reduction in TXNIP, but DASA-58 treatment did (Figure 6).

In summary, our results show that PKM2 is generally expressed in HNSCC, with some cell lines showing higher protein levels and higher levels of the P-PKM2 Tyr105 variant as well, as indicated by Western blot results. Higher abundance of this variant could be explained by an enhanced activity of EGFR or HER2 signaling, as inhibiting these signaling pathways could reduce its levels. Phosphorylation of PKM2 at this residue leads to the dissociation of the active tetramers into dimers or monomers. These are not able to convert PEP to pyruvate and are reported to fulfill functions enhancing the tumorigenicity of the cells.

Although all investigated HNSCC cell lines showed IC_50_ values for Compound 3k in the same micro molar range, not all reacted with an equally strong metabolic response to this PKM2 inhibitor. Only two lines, FaDu and SCC-9, showed a significant reduction in basal and compensatory glycolysis as well. DASA-58 treatment, on the other hand, exhibited no effect on the viability of the cells, but led to an increase in basal and compensatory glycolysis in several HNSCC lines (BHY, SCC-154, SCC-9).

## 3. Discussion

One of the hallmarks of cancer is the strong reliance of tumor cells on glycolysis for covering their energy demands and their need for cellular building blocks. This partial abandonment of the more efficient alternative OxPhos distinguishes tumor cells from their normal counterparts. Over the last few years, it emerged that the glycolytic enzyme PKM2 plays a major role in inducing this metabolic shift [13,14]. Several mechanisms that regulate its conventional glycolytic activity were uncovered and shown to mediate other functions for the enzyme. The aim of our study was to look at some of these influences and modifications in the context of HNSCC.

We could show that in all of our HNSCC cell lines PKM2 is expressed at levels comparable to or even slightly higher than in the cervix carcinoma cell line HeLa. A study in 2014 investigated PKM2 expression in several cancer types, using cancer genome atlas (TCGA) RNA-sequencing data [40]. PKM2 was found to be more highly expressed in HNSCC compared to normal tissue. Furthermore, analyzing patient data available in TCGA, HNSCC samples with high levels of PKM2 were found to correlate with poor prognosis. Accordingly, in 2015 Wang et al. showed that PKM2 is overexpressed in oral squamous cell carcinoma (OSCC), a subtype of HNSCC, and that these patients showed a reduced overall and disease-free survival [41]. This underlines the importance of PKM2 for HNSCC prognosis and shows its potential as a therapeutic target.

All HNSCC cell lines that we looked at showed expression of the Tyr105 variant of PKM2. An investigation in breast cancer cells elucidated that P-PKM2 Tyr105 activates the expression of genes responsible for cancer stemness and enhanced the cancer stem-like cell population [27]. Cancer stem cells are self-renewable with high proliferation and migration capacity and are able to differentiate. They are not only able to maintain these properties in their descendants, but also produce cells without this stemness, which form the bulk of the tumor [42]. An established marker for cancer stem cells is the CD44 receptor. Its expression strongly correlated with P-PKM2 Tyr105 levels in breast cancer cells; P-PKM2 Tyr105 induced translocation of transcription regulator YAP (YES associated protein) to the nucleus, thereby enhancing CD44 expression [27]. In an earlier study in colorectal, glioma and lung carcinoma cell lines, CD44 and PKM2 were shown to directly interact, and CD44 ablation suppressed Tyr105 phosphorylation, thereby increasing PKM2 activity [43]. In our investigation, we observed a higher expression of CD44 in cell lines with high PKM2 and P-PKM2 Tyr105 levels, indicating a connection between PKM2 and cancer stemness in HNSCC as well. The cell line Detroit 562 showed the highest levels of PKM2 phosphorylated at Tyr105, corresponding with the highest expression of the cancer stem cell marker CD44. For metastasis, epithelial cancer cells have to acquire mesenchymal traits, such as the enhanced ability to migrate and to invade other tissues, a process called epithelial–mesenchymal transition (EMT). It was shown that tumor cells that underwent EMT acquire properties of cancer stem cells [44]. Consistent with this finding, the Detroit 562 cell line is derived from a long-distance metastasis of a pharynx tumor [33].

Already in the initial publication describing the inhibiting effect of Tyr105 phosphorylation, different receptor tyrosine kinases (RTKs) were found to be responsible for this modification [26]. An FGFR1 kinase dead mutant was unable to phosphorylate PKM2 in vitro, in contrast to the functional RTK. In vivo data confirmed the importance of FGFR1. Furthermore, different tyrosine kinase inhibitors (TKIs) were tested on a range of leukemic cell lines, implicating that PDGF, BCR-Abl, JAK2, EGFR, FLT3 and cKit can be the phosphorylating kinases as well [26]. In a later publication using a tyrosine antibody array, several RTKs able to bind to PKM2 were detected, and their ability to phosphorylate PKM2 was confirmed when overexpressed in a breast cancer cell line (Axl, EpHA2, FAK, Tyro3, ErbB2, 29440169 [27]). Furthermore, treatment with ErbB2- and EGFR-specific TKI lapatinib reduced P-PKM2 Tyr105 levels in a breast cancer mouse model [27]. We could confirm the effect of lapatinib in HNSCC cell line Detroit 562. Showing high amounts of the phosphorylated PKM2 variant, levels decreased when cells were treated with lapatinib. Furthermore, treatment with EGF could enhance PKM2 variant levels. As this cell line shows high overexpression of EGFR and only moderate expression of ErbB2 (mRNA sequencing data, [37]), this indicates that EGFR activation might be responsible for the high levels of P-PKM2 Tyr105 we observed. Although Detroit 562 cells showed high expression of the HGF receptor Met, and stimulation with HGF was shown to have an impact on glycolysis in our recent publication [37], we could not find a significant influence of this signaling pathway on PKM2 expression or its phosphorylation at Tyr105.

The role of PKM2 activity in cancer progression is not completely clear [45]. There are studies with contradictory findings concerning the right strategy for using PKM2 as a therapeutic target. DASA-58 is a PKM2 activator favoring the formation of active PKM2 tetramers [29]. In the presence of DASA-58, H1299 cells, a non-small cell lung carcinoma cell line, showed reduced secretion of lactate, indicating a lower rate of glycolysis. When H1299 xenografts on nude mice were treated with TEPP-46, an activator comparable to DASA-58, they exhibited a higher percentage of tumor-free injection sites and smaller tumor weights compared to vehicle treatment alone [29]. In contrast, in another study, breast cancer cell lines showed enhanced lactate production and lower oxygen consumption upon DASA-58 treatment. Furthermore, these cells showed higher levels of AMPK phosphorylated at Thr172 in combination with a decrease in TXNIP levels [38].

In the HNSCC cell line BHY, we could confirm the findings, which implicate that DASA-58 acts by enhancing aerobic glycolysis even further rather than inhibiting it [38]. BHY cells reacted to DASA-58 with enhanced basal and compensatory glycolysis, and the oxygen consumption rate was lower compared to control. Levels of phosphorylated AMPK were higher and TXNIP levels decreased. Normally, AMPK is active in response to energy crises accompanying low AMP/ADP to ATP ratios. Its action enhances metabolism in favor of ATP production, similarly to the uptake of glucose, and hinders anabolic processes [46]. TXNIP, on the other hand, reduces uptake and breakdown of glucose [47]. Thus, through its influence on AMPK and TXNIP, DASA-58 treatment can further accelerate glucose metabolism in breast cancer and HNSCC in combination with lowering OxPhos even further. This can lead to a depletion in upstream glycolytic intermediates and can make cells more prone to other stressors attacking, e.g., residual oxidative phosphorylation or the pentose phosphate pathway [38].

Compound 3k treatment resulted in cytotoxicity in all investigated HNSCC cell lines we investigated. Intriguingly, on the metabolic level, only two cell lines with high glycolytic activity showed a strong response. Therefore, it seems that Compound 3k can induce cytotoxicity in HNSCC cells without affecting their glycolysis rate or oxygen consumption. FaDu and SCC-9 reacted on treatment with lower levels of TXNIP, but AMPK phosphorylation at Thr172 remained stable. In contrast, although showing reduced glycolysis rates as well, in a recent publication an ovarian cancer cell line exhibited higher levels of the phosphorylated AMPK variant upon Compound 3k treatment [30]. The low levels of TXNIP in Compound 3k-treated HNSCC lines FaDu and SCC-9 point towards an enhanced glucose uptake, but compared to untreated cells, more of the glucose is shuttled into oxidative phosphorylation, as indicated by the higher consumption of oxygen. It is not known how Compound 3k and other PKM2 inhibitors are able to inhibit PKM2 on the molecular level. Binding of the allosteric activator FBP induces the formation of the active FBP tetramer. It was shown for two other inhibitors that FBP does not restore activity of inhibited PKM2, indicating that their point of action might be the FBP binding site [48,49].

In summary, our findings show that in HNSCC, a prominent amount of PKM2 can already be inactive due to posttranslational modification at tyrosine 105. This correlates with the expression of CD44, a well-defined marker for cancer stemness. Treatment of HNSCC with TKIs such as lapatinib could inhibit phosphorylation and thereby impede the protumoral effects of the phosphorylated variant.

Due to the heterogeneous results of our studies with the PKM2 inhibitor and activator, it is not possible to deduce a preferable strategy for PKM2-targeted HNSCC treatment. The inhibitor Compound 3k induced cytotoxicity on HNSCC cells, but it is unclear if this is really based on a manipulation of metabolism. Only cell lines with high glycolytic activity showed a metabolic response. PKM2 activation by DASA-58 led to higher glycolysis and lower oxygen consumption and showed no cytotoxic effect. Although this might be at first sight contradictory, an even higher glycolysis could lead to a strongly impaired metabolism due to lack of intermediates of glycolysis. Activators that are more efficient or co-treatment with inhibitors of other metabolic pathways could be an option to further investigate PKM2 activation in HNSCC.

## 4. Materials and Methods

### 4.1. Cell Lines

Detroit 562, FaDu and SCC-9 were obtained from ATCC (Manassas, VA, USA) and cultured as described elsewhere [37]. SCC-154, HN and BHY were obtained from DSMZ (Braunschweig, Germany). SCC-154 was cultured in MEM (with Earle’s salts) with additional 1% of non-essential amino acids; HN and BHY were cultured in DMEM (all media obtained from Thermo Fisher Scientific, Waltham, MA, USA). Cell culture media were supplemented with 10% FCS and 1% Pen/Strep (Thermo Fisher Scientific, Waltham, MA, USA).

### 4.2. Western Blotting

For analyzing the amount of PKM2 and its phosphorylated variant (P-PKM2 Tyr105), 250,000 cells were seeded in 12-well plates. Cells were prepared for Western blot analysis after 48 h, as described elsewhere [37].

For investigating the effects of growth factors, 250,000 cells were seeded in 12-well plates. On the next day, they were treated with vehicle alone, the growth factor alone or the growth factor in combination with a suitable inhibitor. To another well, the inhibitor alone was added. Cells were incubated for 48 h and prepared for Western blot analysis as described elsewhere [37]. FGF2 (PHG0266) and HGF (PHG0254) were obtained from Thermo Fisher Scientific (Waltham, MA, USA), and EGF (E9644) from Merck (Darmstadt, Germany). Appropriate inhibitors AZD4547, foretinib and lapatinib were obtained from Selleck Chemicals (Houston, TX, USA).

After gels were run and blotted, nitrocellulose membranes were blocked with 5% dry milk in TBS for 1 h. Blots were incubated overnight at 4 °C with primary antibodies (PKM2 # 4053, P-PKM2 Tyr105 #3827, Vinculin # 13901, TXNIP #14715, P-AMPKα Thr172 # 2535, AMPKα #2532, Cell Signaling, Danvers, MA, USA) diluted according to the instructions of the manufacturer. The next day, blots were washed and incubated with the appropriate HRP-coupled secondary antibody for 1 h, and after washing were subjected to signal detection (ECL Western Substrate, Thermo Fisher Scientific, Waltham, MA, USA; ChemiDoc Imaging System, Bio-Rad, Hercules, CA, USA).

Protein bands were analyzed for quantification using Image Lab Software 6.0.1 (Bio-Rad, Hercules, CA, USA). Bands were normalized by dividing intensities of bands of proteins of interest with the intensities of the corresponding Vinculin bands (ratio values depicted directly in the blot). Alternatively, band intensities were normalized by dividing all samples of the same cell line on the same blot with the value of the sample showing the highest intensity. Intensities of the proteins of interest were then adjusted by dividing them with the corresponding Vinculin ratio value. For presenting a bar chart combining several independent experiments, results were presented in relation to an internal control (vehicle-treated control or control cell line HeLa).

### 4.3. Cytotoxicity Assay

Ten thousand cells per well were seeded in 96-well plates and on the next day were kept untreated or were treated in duplicate with 2-fold decreasing concentrations of Compound 3k and DASA-58 (Selleck Chemicals, Houston, TX, USA) dissolved in DMSO starting at 200 µM. For the controls, cells were also treated in duplicate with corresponding amounts of DMSO alone. After 48 h, the cells were incubated for 10 min with 50 µL 0.5% (*w/v*) crystal violet solution and washed three times using distilled water. To air-dried plates, 100 µL of 98% methanol per well was added, and absorption at 595 nm was measured (Photometer Infinite F50, Tecan, Männedorf, Switzerland). For curve fitting, data were normalized on a scale of 0–100% viability and analyzed with the software Prism (GraphPad, San Diego, CA, USA) using nonlinear regression, log (inhibitor) vs. normalized response.

### 4.4. Metabolic Measurements

Glycolytic rate assays were performed using a Seahorse XF 96 analyzer (Agilent, Santa Clara, CA, USA) for measuring extracellular acidification rate (ECAR) and oxidative consumption rate (OCR). Cells were seeded in 96-well XF culture plates (Agilent, Santa Clara, CA, USA) in assay medium and were incubated with 30 µM Compound 3k, DASA-58 (Selleck Chemicals, Houston, TX, USA) or DMSO alone (10 replicates for each condition) for 5 or 16 h. After three basal measurements of OCR and ECAR (for basal glycolysis), 5 µM of Rotenone and Antimycin A (Merck, Darmstadt, Germany) were injected for inhibition of complex I and complex III of the electron transport chain of OxPhos. Three measurements followed (for compensatory glycolysis). OCR measurements in combination with ECAR measurements without a functional electron transport chain were used to determine the proton efflux rate solely derived from glycolysis (glycoPER). A concluding injection of 500 mM of glycolysis inhibitor 2-desoxy-glucose (2-DG, Merck, Darmstadt, Germany) was applied and followed by an additional five measurements, for proving that the remaining acidification measured originated from glycolysis (post 2-DG acidification).

For experiments with an incubation time of 30 min, 10 fold concentrated stock solutions of Compound 3k, DASA-58 or DMSO were injected into the plates using the Seahorse XF 96 analyzer after three basal measurements. Five measurements of OCR and ECAR were performed (corresponding to 30 min of incubation time), before Rotenone and Antimycin A were injected. The assay protocol then proceeded as stated for the other two time points.

Subsequent to the measurements, cells were stained with crystal violet solution as described in Section 4.3. Absorption data at 595 nm were used for normalizing for cell count. Analysis was performed using Wave Software and Glycolytic Rate Report Generator (Agilent, Santa Clara, CA, USA).

### 4.5. Experimental Design and Statistics

Independent experiments were performed at least three times, and mean values of experiments with SD values are presented. If only one representative experiment without a summary graph is shown, it is indicated in the figure caption. For quantification of Western blots, detection of a household protein was used to compensate for unequal loading (see Section 4.2. for details). Metabolic measurements were performed with 10 replicates per experiment, and independent experiments were performed after three different incubation times (30 min, 5 h and 16 h). Outcomes of all three experiments were used to rank the susceptibility of the different cell lines to the tested reagents (Table 1 and Table 2). For statistics, the program Prism (GraphPad, San Diego, CA, USA) was used. Data are presented in relation to a control that was set to one. Therefore, *p*-values were calculated using the one sample t-test function with a hypothetical value of one.

## Figures and Tables

**Figure 1 ijms-23-00775-f001:**
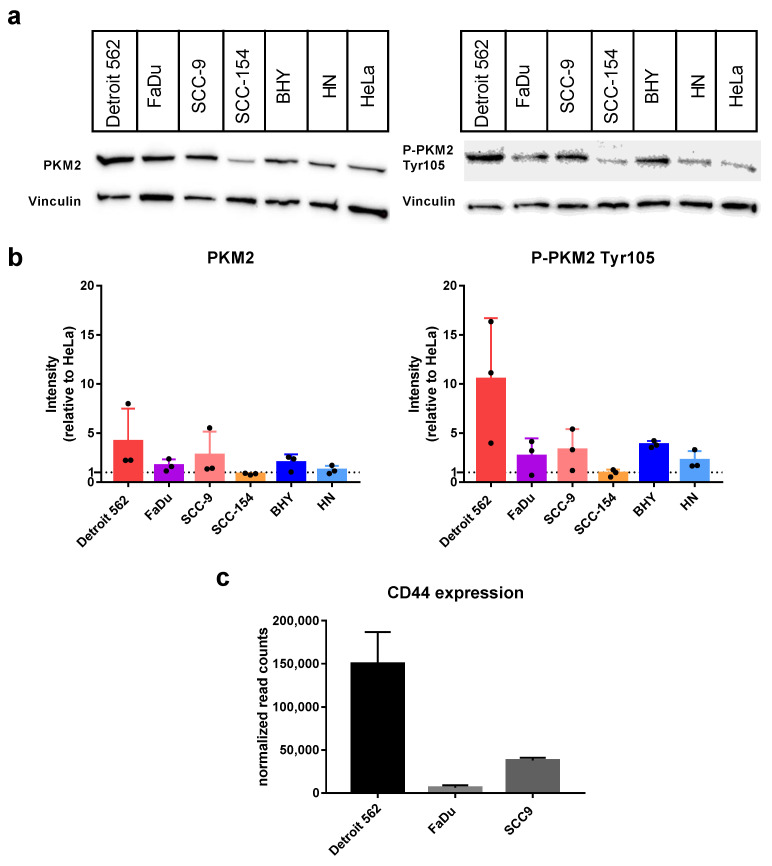
Abundance of PKM2 and its variant P-PKM2 Tyr105 in HNSCC cell lines. (**a**) Western blots of indicated cell lysates with an antibody specific for PKM2 (left) or P-PKM2 Tyr105 (right). For controlling equal loading, upper parts of blots were incubated with an antibody specific for Vinculin. (**b**) Graphs displaying Western-blot intensities relative to result for Hela reference cell line. Shown are mean values and SD of four independent experiments as depicted in (**a**). Vinculin intensities were used for normalization. (**c**) Normalized expression levels of the gene CD44 obtained by mRNA sequencing (see [37] for details on sequencing).

**Figure 2 ijms-23-00775-f002:**
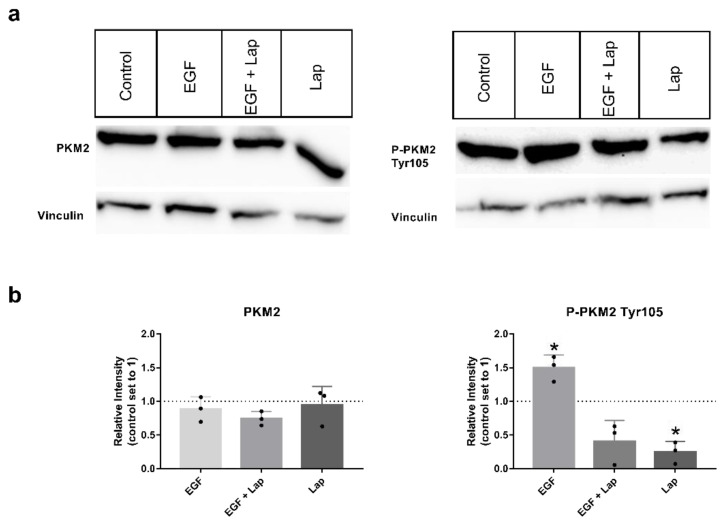
EGF stimulation and lapatinib treatment have an impact on levels of P-PKM2 (Tyr105). (**a**) PKM2 (**left**) or P-PKM2 Tyr105 (**right**) specific Western blots of Detroit 562 cells stimulated for 48 h with 8.3 nM EGF, 8.3 nM EGF in presence of 10 µM lapatinib (EGF + Lap), 10 µM lapatinib only (Lap) or with vehicle alone (Control). Vinculin was used as a loading control. (**b**) Western blot intensities relative to untreated control. Shown are mean values and *SD* of three independent experiments as depicted in (**a**). Vinculin intensities were used for normalization. *: *p* < 0.05, one sample *t*-test.

**Figure 3 ijms-23-00775-f003:**
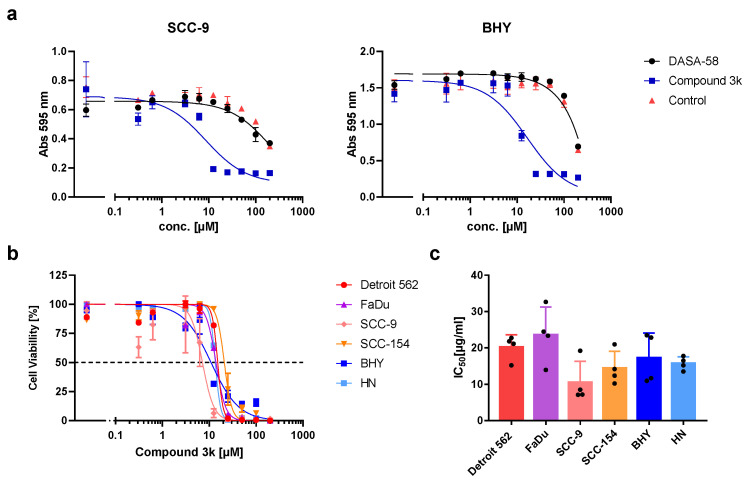
Cytotoxic activity of DASA-58 and Compound 3k on HNSCC cell lines. (**a**) Viability of cell lines SCC-9 and BHY treated with different concentrations of indicated compound or DMSO (control) for 48 h. Cells were stained with crystal violet, and absorption at 595 nm was measured. First data points on the left correspond to cells treated with medium alone. (**b**) Normalized cell viability after treatment with different concentrations of Compound 3k for 48 h. Data were fitted using non-linear regression. Dotted line indicates 50% cell viability. (**c**) IC_50_ values (inhibitor concentration at 50% viability) for Compound 3k derived from data as shown in (**b**). Mean values and SD of four independent experiments are depicted.

**Figure 4 ijms-23-00775-f004:**
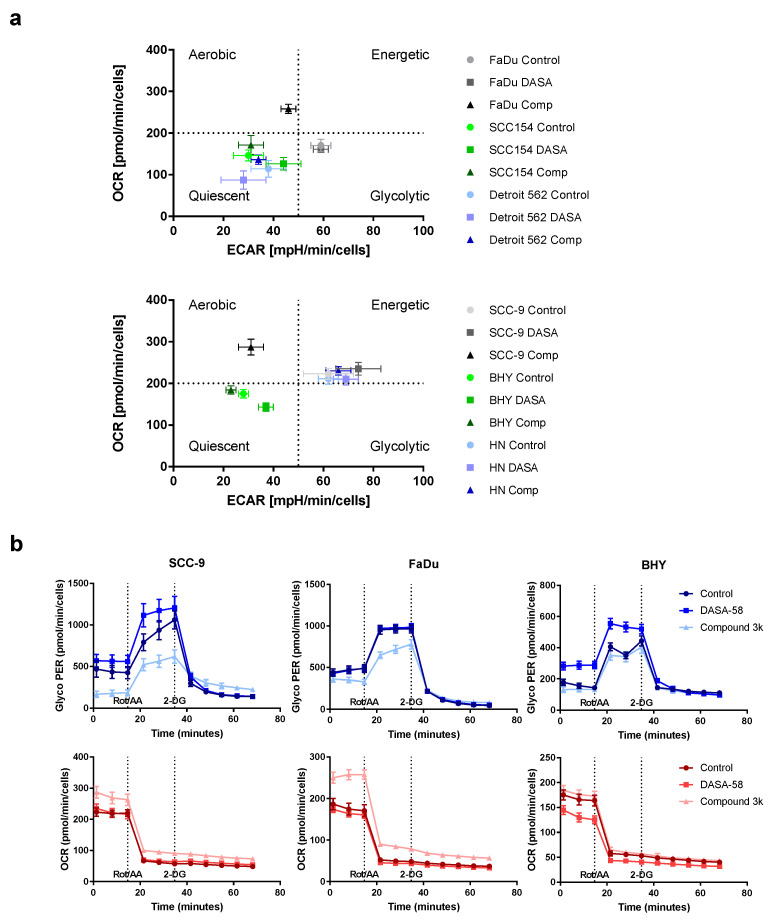
Influence of DASA-58 and Compound 3k on HNSCC metabolism. (**a**) Indicated HNSCC cell lines were treated for 5 h with 30 µM Compound 3k (Comp), 30 µM DASA-58 (DASA) or with vehicle alone (Control) and were subsequently measured in a real-time cell metabolic analyzer. Basal ECAR values were plotted against the corresponding OCR values. Graphs are divided into four areas (dotted lines) to define different metabolic groups. (**b**) Representative selection of results of glycolytic rate assays performed with different HNSCC cell lines after 5 h of stimulation with Compound 3k, DASA-58 or with vehicle alone (Control). GlycoPER kinetic graphs in blue, corresponding OCR kinetic graphs below in red. Dotted lines indicate time points of injections of Rotenone in combination with Antimycin A (Rot/AA) and 2-Desoxy-D-Glucose (2-DG).

**Figure 5 ijms-23-00775-f005:**
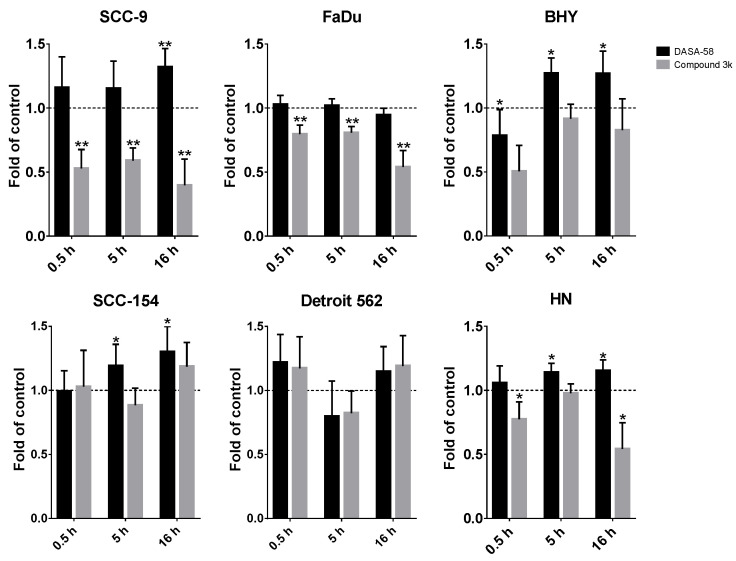
DASA-58 and Compound 3k can affect compensatory glycolysis. Compensatory glycolysis (corresponds to the GlycoPER value of measurement 6 in the kinetic graphs as shown in Figure 4b, Appendix A, and measurement 11 for 0.5 h stimulation in Appendix A) after 0.5, 5 h and 16 h of treatment with 30 µM Compound 3k or DASA-58 in all six investigated HNSCC cell lines. Shown are changes in relation to the vehicle-treated control. Data points represent means with SD, *n* = 10. *: *p* < 0.01, **: *p* < 0.001, one sample *t*-test.

**Figure 6 ijms-23-00775-f006:**
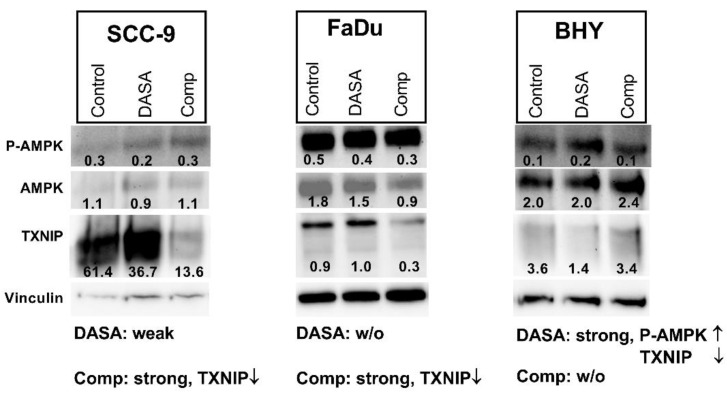
Influence on P-AMPK and TXNIP levels. Western blot of cell lysates after 24 h of treatment with 10 µM DASA-58 (DASA), 10 µM Compound 3k (Comp) or with vehicle only (Control). Numbers indicate ratios of band intensity to intensity of corresponding Vinculin band. The inscription below outlines the metabolic effects of the substances as given in Table 1 and Table 2 (weak, strong, *w/o*: without effect) combined with observed changes in depicted Western blots (↑: intensity is increased, ↓: intensity is decreased upon treatment). Shown is one representative result out of three.

**Table 1 ijms-23-00775-t001:** Outcomes of DASA-58 treatment on HNSCC cell lines. o: no effect, #: small effect, ##: large effect. Five or more times small or large effect results in total effect “strong”, two to four times small or large effect results in total effect “weak”, zero to one time small or large effects results in total effect “*w/o*” (without effect). Comp.: Compensatory.

DASA-58		SCC-9	HN	FaDu	SCC-154	Detroit 562	BHY
type		energetic	energetic	glycolytic	quiescent	quiescent	quiescent
ECAR/OCR	5 h	o	o	o	##	o	##
Basal Glycolysis	0.5 h	o	#	o	o	o	o
	5 h	o	##	o	##	o	##
	16 h	##	##	o	##	o	##
Comp. Glycolysis	0.5 h	o	o	o	o	o	#
	5 h	o	#	o	#	o	#
	16 h	##	#	o	#	o	#
**Total effect**		weak	strong	*w/o*	strong	*w/o*	strong

**Table 2 ijms-23-00775-t002:** Outcomes of Compound 3k treatment on HNSCC cell lines. o: no effect, #: small effect, ##: large effect. Five or more times small or large effect results in total effect “strong”, two to four times small or large effect results in total effect “weak”, zero to one time small or large effects results in total effect “*w/o*” (without effect). Comp.: Compensatory.

Comp-3k		SCC-9	HN	FaDu	SCC-154	Detroit 562	BHY
type		energetic	energetic	glycolytic	quiescent	quiescent	quiescent
ECAR/OCR	5 h	##	o	##	o	o	o
Basal Glycolysis	0.5 h	o	o	#	o	o	#
	5 h	##	o	##	o	o	o
	16 h	o	o	##	o	o	o
Comp. Glycolysis	0.5 h	##	o	##	o	o	o
	5 h	##	#	##	o	o	o
	16 h	##	#	##	o	o	o
**Total effect**		strong	weak	strong	*w/o*	*w/o*	*w/o*

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
