# Peer review of "PKM2 Modulation in Head and Neck Squamous Cell Carcinoma"

_ijms, 2022, doi:10.3390/ijms23020775_

Round 1

Reviewer 1 Report

An interesting original article exploring the role of enzyme pyruvate kinase M2 in head and neck squamous cell carcinoma, showing that this kinase is well expressed in this type of tumor also indicating a role in cancer stemness; only minor revisions:

A statistical analysis subsection, better describing the statistical program used, the various test would be helpful

Page 2 line 59 I think that a clinical description of head and neck squamous cell carcinoma is necessary; the authors could add a sentence such as: "Head and neck cancer is a term used to define malignant tumors developing on the lips, mouth, nose, and other head and neck areas. The majority of these tumors are squamous cell carcinomas, a tumor characterized by an abnormal and quick growth of keratinocytes in the epidermis" and a citation such as:  doi: 10.3390/curroncol28040213. and doi: 10.3390/medicina57060563.

Thank you

Author Response

Dear Reviewer,

thank you very much for reading our manuscript and your helpful suggestions. See below our changes in the revised manuscript.

As you recommended we added a paragraph to the materials and method section containing information on experimental design and statistics (beginning at line 482). In the materials and methods section 4.2. we now included a more detailed description of Western Blot quantification (line 436).

Thank you very much for your suggestion to add a clinical description of HNSCC directly at its first mentioning in the introduction. We changed our text at line 59 and put in a new citation describing treatment options of HNSCC (citation 23).

In addition, resolution of Western Blot pictures was enhanced in figure 1 and 2. Also, we described in more detail why we used HeLa as control for the blots in figure 1 (see line 119).

Thank you very much for your review,

the authors

Reviewer 2 Report

We all know that PKM2 plays a major role in the switch of tumor cells from oxidative phosphorylation to aerobic glycolysis, The author confirmed that activating or interfering with the activity of PKM2 by drugs can significantly affect the aerobic glycolysis capacity of tumor cells in HNSCC, this may provide new ideas for the clinical treatment of NHSCC. On the other hand, the authors also found that different HNSCC cell lines respond differently to PKM2 activators or inhibitors.

However, the author has not verified the effect of PKM2 activators or inhibitors on the survival of HNSCC cells in vitro or in vivo. I think that the result of an E-CAR experiment is too far for clinical transformation.

Here are some specific comments:

  1. In FIG. 1a, the picture quality of the WB experiment is too low, and Why use Hela cells instead of the normal cervical squamous epithelium as a control.
  2. In FIG. 2a, the picture quality of the WB experiment is too low.

Author Response

Dear Reviewer,

thank you very much for reading our manuscript and your helpful suggestions. See below our changes in the revised manuscript.

We apologize for the low resolution of western blot pictures in figures 1 and 2. We changed them accordingly. Residual blurriness of P-PKM2 Tyr 105 western blots is a feature of the antibody and probably also the lower abundance of the phosphorylated variant; signal intensity is significantly lower compared to total PKM2.

HeLa has been used as positive control as it is described to express PKM2. Apart from that, having this sample on all our blots makes it possible to express the signals obtained in our HNSCC cell lines as relative numbers and to blot graphs which summarize several experiments as shown in figures 1b. In our text we now describe in more detail why we used HeLa as control for the blots in figure 1 (see line 119). Furthermore, in the materials and methods section 4.2. we now included a more detailed description of Western Blot quantification (line 436).

Concerning your suggestion that we should include data showing the effect of PKM2 activation and inhibition in vitro or in vivo we want to point out that we had already included in vitro data in the first version of our manuscript (see figure 3). Furthermore in our manuscript we did not imply that PKM2 activators or inhibitors are in any way near to clinical transformation. Our manuscript was inspired by recent publications showing the effects of DASA-58 and Compound 3k in other tumor entities, publications we thoroughly discuss in our manuscript and we compared our findings in HNSCC with (citations 30 and 38).

When looking at the cytotoxicity data in figure 3 one can see that DASA-58 had no impact on the cells beyond the effect of DMSO control. We showed curves for SCC-9 and BHY only, but stated that it’s the same for the other lines (see line 169). Compound 3k exhibited a rather high IC50 -value and activity did not correspond with an effect on metabolism (see also lines 377 onward in discussion).

Other changes to the revised manuscript are that we slightly rearranged the introduction part explaining head and neck cancer and included another citation (line 59 onwards, citation 23) for better understanding. We also added a new section in materials and methods describing experiment design and statistics (starting at line 482).

Thank you very much for your review,

the authors